# Nanomechanical and Vibrational Signature of Chikungunya Viral Particles

**DOI:** 10.3390/v14122821

**Published:** 2022-12-17

**Authors:** Ruana Cardoso-Lima, Joel Félix Silva Diniz Filho, Maria Luiza de Araujo Dorneles, Renato Simões Gaspar, Pedro Filho Noronha Souza, Clenilton Costa dos Santos, Daniela Santoro Rosa, Ralph Santos-Oliveira, Luciana Magalhães Rebelo Alencar

**Affiliations:** 1Laboratory of Biophysics and Nanosystems, Physics Department, Federal University of Maranhão, Maranhão 65080805, Brazil; 2Vascular Biology Laboratory, Heart Institute (InCor), University of Sao Paulo School of Medicine, São Paulo 05468000, Brazil; 3Department of Biochemistry, Federal University of Ceará, Ceará 60430275, Brazil; 4Drug Research and Development Center, Department of Physiology and Pharmacology, Federal University of Ceará, Ceará 60430275, Brazil; 5Department of Microbiology, Immunology, and Parasitology, Federal University of São Paulo, São Paulo 04023062, Brazil; 6Brazilian Nuclear Energy Commission, Nuclear Engineering Institute, Rio de Janeiro 21941906, Brazil; 7Laboratory of Nanoradiopharmacy, Rio de Janeiro State University, Rio de Janeiro 23070200, Brazil

**Keywords:** CHIKV, physical virology, ultrastructure, AFM, Raman

## Abstract

Chikungunya virus (CHIKV) belongs to the genus Alphaviridae, with a single-stranded positive-sense RNA genome of 11.8 kbp encoding a polyprotein that generates both non-structural proteins and structural proteins. The virus is transmitted by the Aedes aegypti and A. albopictus mosquitoes, depending on the location. CHIKV infection leads to dengue-like musculoskeletal symptoms and has been responsible for several outbreaks worldwide since its discovery in 1952. Patients often experience fever, headache, muscle pain, joint swelling, and skin rashes. However, the ultrastructural and mechanical properties of CHIKV have not been fully characterized. Thus, this study aims to apply a physical approach to investigate CHIKV′s ultrastructural morphology and mechanical properties, using atomic force microscopy and Raman spectroscopy as the main tools. Using nanomechanical assays of AFM and a gold nanoparticles substrate for Raman signal enhancement, we explored the conformational plasticity, morphology, vibrational signature, and nanomechanical properties of the chikungunya virus, providing new information on its ultrastructure at the nanoscale and offering a novel understanding of the virus’ behavior upon mechanical disruptions besides its molecular composition.

## 1. Introduction

Chikungunya virus (CHIKV) is a single-stranded RNA alphavirus with a reported diameter of approximately 70 nm [1]. These viruses have a monopartite positive-sense RNA genome enclosed in an icosahedral symmetry capsid surrounded by an envelope with spicules on the surface. Such spicules comprise heterodimers of E1 and E2 glycoproteins, facilitating attachment to cell surfaces [2,3,4]. CHIKV infections have been responsible for several outbreaks worldwide since their discovery [5]. Its transmission occurs mainly through the bite of previously infected female A. albopictus or A. aegypti mosquitoes [6,7]. CHIKV represents a significant threat because it causes abrupt (acute) and prolonged (chronic) symptoms of polyarthralgia and myalgia, which tend to be much more severe than other arboviruses. Importantly, patients might experience long-term sequelae of unclear pathophysiological explanations [7,8]. 

Structural investigation of viruses supported by high-resolution microscopy techniques has been a breakthrough in uncovering new aspects of viral structural components and functions. Nevertheless, there are complexities related to the dynamic nature of viruses that do not fit into static glimpses [9]. Still, significant advances can be achieved using different approaches to broaden information about a particular virus [10,11]. The transition of viral particles from immature to mature particles, entry into the host cell through interaction with the cell membrane, the replication process, movement during infection, and evasion of the immune system are all processes closely associated with the physical properties of a viral particle [12,13,14,15].

Therefore, studying mechanical properties could clarify the relationship between the structure and function of viral particles. Atomic force microscopy (AFM) is a powerful technique that can be used to investigate these mechanical properties at the nanoscale [13,16] and, together with Raman spectroscopy [17,18] analysis, can be employed to obtain the fingerprint of molecular components. Scrutiny of mechanical characteristics such as stiffness, elasticity, and adhesion of viruses, in addition to possible points of fatigue and breakage of the viral capsid [10,11,19,20,21], could provide structural properties unique to different virus particles. In this study, we aimed to characterize the mechanical properties of CHIKV and map its ultrastructure using atomic force microscopy. In addition, we intended to design a methodological approach to perform label-free surface-enhanced Raman spectroscopy to record the vibrational signature of the CHIKV particle.

## 2. Materials and Methods

### 2.1. Virus Culture and Inactivation

A Brazilian isolate of the CHIKV ECSA strain (Genbank: KP164569) was propagated in Vero E6 cells (ATCC, CRL-1586) in MEM medium (Gibco) (supplemented with 10% fetal bovine serum (Gibco) and 1% (*v*/*v*) penicillin/streptomycin (Gibco)-M10) for 48 h. Then, the supernatant of infected cells was collected, harvested, and titrated as previously described. The virus was inactivated by thermal treatment at 56 °C for 1 h [22].

### 2.2. Atomic Force Microscopy (AFM)

AFM measurements were performed according to the methodology employed by Cardoso-Lima et al. [10], where 3 µL of solution with viral particle suspensions were deposited on glass slides (13 mm diameter). The slides were analyzed by a Multimode 8 (Bruker, Santa Barbara, CA, USA), and the probes used were SNL (Bruker) with a 0.24 N/m nominal spring constant and a nominal tip radius of 2 nm in the peak force quantitative nanomechanics (QNM) mode. Viral particle indentation experiments were performed on six viral particles, each undergoing 30 to 70 indentation cycles. For the indentation analysis, measurements were performed on the QNM Ramp Mode following the same procedure used by Cardoso-Lima et al. [10]. We applied a force setpoint of 25 nN and a tip velocity of 100 nm/s. AFM data were analyzed, and the maps were obtained using Nanoscope Analysis software.

### 2.3. Raman Spectroscopy (RS)

The Raman scattering measurements were performed on a micro-Raman system, model T64000 (Horiba/Jobin-Yvon), operating in the single mode. A 2 mW diode laser operating at 785 nm was used as the excitation source. A neutral density filter (1%) was used to avoid laser-induced damage to the sample. The light was focused on the sample using a microscope model BX41 (Olympus), with a 100× objective lens (NA = 0.9, WD = 0.21 mm), and the Raman signal was dispersed in an 1800 gr/mm grid and detected in a liquid-nitrogen-cooled CCD. The slits of the spectrometer were adjusted to obtain a spectral resolution of 2 cm^−1^. A substrate with gold nanorods deposited was used for surface-enhanced Raman scattering (SERS). The spectrum was acquired after three acquisitions of 30 s in each dispersion band of the spectral grid [23,24].

## 3. Results and Discussion

This study investigated the plasticity, morphology, and adhesion properties of CHIKV providing new information on its ultrastructure at the nanoscale and offering an understanding of the virus’ behavior upon mechanical disruptions. The high-resolution topographic maps revealed the structures of the viral surface and its protein distribution (Figure 1A). It is possible to observe the dispersion and calculate the size of the particles, which were 52.04 nm ± 7.6 nm (Figure 1A,C). The topographic analysis revealed that the viral particle has an icosahedral shape, with surface bumps due to protein organization on the particle’s surface (E1 and E2 glycoproteins). This fact is exemplified in Figure 1D, with a superposition of a simulated model of protein organization [25] over the topographic map.

Viral particles are not rigid solids, although CHIKV mainly comprises closely packed glycoproteins that form the icosahedral capsid. As an enveloped virus, there is a lipidic layer involving the protein capsid in which the surface proteins are anchored [26]. Physically, this means that the protein capsid is enclosed by a lipidic outer layer that will fall over, underlining the particle’s internal, more rigid structures [27,28]. It was possible to observe the triangular organization of proteins that form the capsid characteristic of icosahedral viruses. In Figure 2A, we demonstrate the topographic map of two adsorbed particles highlighting the triangular shape of the upward viral particles, while in Figure 2B, we depict their respective 3D map.

Figure 3A shows a viral particle before the indentation cycles with its corresponding adhesion forces map shown in Figure 3B. Interestingly, we could observe the small triangle facet on the top of the virion (Figure 3B, purple arrow). In addition to topographical characterization, nanoindentation experiments were performed to evaluate the stiffness and resistance of the viral particles. These experiments revealed that CHIKV particles could withstand a peak force of up to 25 nN, which is higher than what is supported by ZIKV [11] and SARS-CoV-2 [10]. After 30 loading cycles of indentation, CHIKV particles did not collapse, suffering only local damage (Figure 3C). This fact could be related to the nature of the protein capsid and the tight package of the proteins composing the structural CHIKV particles [26]. This structure could confer a high mechanical resistance and stability to the virion, offering a more plastic response to mechanical stimuli rather than a more compliant one. During infection, viral particles are subjected to forces exerted by different target cells and tissues [29]. Ozden and collaborators [30] showed that CHIKV replicates in the muscle tissue cells of infected subjects. Muscle tissue is extremely resistant, and in this regard, CHIKV particles have an advantage: high mechanical resistance to local disturbances. The functional relevance of these mechanical properties should be explored in the future.

For the adhesion map, the lighter regions (Figure 3B,D) are associated with a positive charge distribution on the particle’s surface [31], using the AFM probe as a reference. It is possible to observe contrasts in force related to the different protein domains (E1 and E2 glycoproteins) of the CHIKV virion. Correlating the topography and adhesion maps in Figure 3D, it is possible to observe changes in the charge distribution of the forces on the viral surface caused by the indentations performed. Such indentations caused a disturbance in the protein organization in these regions, exposing internal portions of the particle.

Adhesion maps are related to the probe and sample surface interaction forces. It could also detect interactions between the different structures of the particle itself. These adhesive interactions are electrostatic, van der Waals and capillary forces, and forces promoted by chemical bond breakage [31]. Usually, AFM results of adhesion are treated as non-specific interactions because of the difficulty in determining the amount of each interaction covered on the measurement, aside from the measurements performed using functionalized probes, which is not the case for the methodology used in this study. The probes we used to analyze all samples have the same composition (Si_3_N_4_) with well-defined specifications, such as geometry and tip radius, and the same conditions for all the assays performed, including temperature and air humidity.

Previous literature [31] on the subject reports that the variations on the distribution of adhesion forces on the maps are mostly associated with van der Waals and electrostatic interactions, in this case especially because we used a non-modified probe in low humidity conditions, with a conical sharp probe, which makes unfeasible the contributions of chemical or capillary forces for the adhesion. Moreover, for the frequency of the peak force measurements (1 kHz) in low humidity air medium, some triboelectrification of the probe is expected because of the absence of charge dissipation. So, the accumulation of charges on the sample surface, related to the sample’s composition, even for non-conductive samples, results from this variance in electrostatic forces between the probe, air, and the sample. Regarding the contribution of van der Waals forces, they have components of orientation, induction, and dispersion forces, the latter being a dipole/dipole-induced interaction that contributes more significantly. The van der Waals forces experienced by the AFM probe depend on the geometric parameters of the tip, in addition to being directly proportional to the Hamaker constant, which includes physicochemical parameters of the probe-sample interaction. Thus, changes in the contributions of van der Waals forces can only be provided by changes in the potential of the probe-sample interaction, and, in this case, it must come from the heterogeneity of the sample, as the probe atoms are always the same for all the measurements [10,32,33,34].

For each indentation cycle, a force curve was obtained as a result of the approximation and retraction movement of the probe relative to the sample. In Figure 3E, we have a representation of a single force curve that shows the rupture events that occurred over this cycle of indentation on the top of the CHIKV virion. In the yellow portion, it is possible to notice two rupture events in the approximation curve (in blue) that were reported by previous literature as signs of breakage caused by mechanical load [13,35]. The first event occurs at about 1.4 nN, and the second occurs around 4.1 nN, indicating that at least two different layers are present and were broken, summing to 10 nm thickness. The negative portion of the retraction curve (in red) is characteristic of adhesive/attractive interactions. Indeed, it is possible to notice a pattern of Velcro-like events [36]. This pattern is observed when protein bonds are broken, confirming the rupture events during the exit of the tip from the virion surface.

Throughout our AFM analyses, we observed the formation of fibers (Figure 4) in different CHIKV samples. These fibers were 561 ± 87 nm in length and 15.68 ± 3.6 nm in diameter and could be RNA- or protein-related formations. We found no literature on similar structures being identified in CHIKV samples. In Figure 4A, there is a 3D height map; in Figure 4B, the cross-section represents the measurement of the diameters, with an inset map showing a dotted line in the exact position of the section. For Figure 4C,D, the maps showcase the mechanical signature of these fibers. The adhesion map in Figure 4C outlines the charge distribution of the fiber compared to the substrate in reference to the tip. The difference in contrasts on the map is a consequence of a greater or lesser accumulation of charge on the surface, which is associated with the fiber’s composition and/or structure [31]. Unlike viral particles, the fibers present a more negative surface charge. Figure 4D shows the energy dissipation map, which is associated with the viscoelastic nature of a solid object. In both mechanical maps, it is possible to observe that the fibers are composed of some repeating building blocks, demonstrating a periodicity size of about 36.47 nm ± 3.03 nm.

As for a reason for the appearance of these fibers, it is possible that some CHIKV particles have collapsed, and the genetic content from inside the capsid has spilled out and formed these fiber-like structures. Another feasible explanation is that, upon collapsing, E1 and E2 glycoproteins in the CHIKV capsid can re-arrange together with the genetic content to form these fiber structures. Literature reports similar structures related to protein self-assembly [37] and some even with RNA and DNA structures [38]. Another possible association with the knowingly negatively charged RNA [39] is the also negative surface charge of the fibers, as demonstrated by the adhesion map in Figure 4C.

It is worth noting that we did not observe the formation of these fiber-like structures when analyzing other viruses [10,11], despite some of these being RNA viruses. Moreover, the experiment was performed multiple times using different slides, and even comparing substrates (mica and glass). The appearance of fiber formation was present in all of them. One key difference between CHIKV and other viruses previously studied by our group is the presence of E1 and E2 glycoproteins in CHIKV. Therefore, it is reasonable to speculate that these glycoproteins may be related to forming these fiber-like structures. Indeed, these fibers resemble collagen fibrils in shape and length [40], while type I collagen was identified as a binding partner of the E2 glycoprotein [41]. However, the functional consequences of such interactions are unclear. Suppose these fiber-like structures are indeed composed of E2 glycoprotein. In that case, CHIKV likely forms these fibers to facilitate the adhesion of collagen-producing cells, which might help to explain why CHIKV infection leads to prominent musculoskeletal symptoms. These CHIKV fibers could also be involved in long-term sequelae and, therefore, will be further characterized in forthcoming studies.

To identify molecular compositions and provide a vibrational signature of CHIKV particles and these newly identified fiber-like structures, we performed SERS measurements. In the spectra obtained (Figure 5), it was possible to locate bands distinguishing RNA nucleic and lipidic groups, amino acids, and others. All four RNA bases had outstanding signals, with a medium-to-strong intensity of the peak for Adenine and Guanine and medium-to-weak for Cytosine and Uracil peaks. The exact wavelength numbers for each of the bases are stated in Table 1.

The two peaks of phosphodiester (O-P-O) around 800 cm-1 are related to the RNA backbone, which links the nuclear bases [46]. In addition, there is the presence of the amino acids that make up the proteins, such as tyrosine (850 cm^−1^), proline (937 cm^−1^), and L-tyrosine (1200 nm^−1^) that can be related to the glycoproteins [43] on the surface of the virion. Moreover, there are bands for polysaccharides (950 cm^−1^) and lipids (1305 cm^−1^ and 1450 cm^−1^) that relate to the composition of the viral membrane [42,44]. In the interval from 1230 cm^−1^ to 1260 cm^−1^, there is the presence of the amide III band assignment. The amide group is associated with the functional group CONH (carbon-oxygen-nitrogen-hydrogen), which essentially creates linkages for protein formation, conferring structural rigidity [45,47]. The amide group can be divided into subgroups referred to as amide A, amide B, and amide I to VII, where amides I, II, and III are of most interest for the documentation of different protein structural conformations. The spectrum of CHIKV presented most prominently the band for the amide III group, which can be comprised of enzymes, antibodies, transport or membrane proteins, and viral coats [47].

The non-appearance of the amide I band is noteworthy in the spectra obtained, which is representative of protein domains in Raman spectroscopy results. This absence is related to the length of the amino acid side chain. The side chains increase the distance between the peptide bond and the metallic nanoparticles, preventing them from coming into direct contact. However, it is also possible that the lack of the viral amide spike is also related to the virus, an inactivation program that breaks down amino acids [50].

It is important to point out that, for label-free SERS measurements, it is virtually impossible to know whether the laser hits a virus particle since the positioning of the laser is performed with the use of an optical microscope that is not even near the necessary resolution to spot a viral particle. A solution for this could be using a tip-enhanced Raman spectroscopy (TERS) measurement as performed by Dou and collaborators [51], where the probe can give the exact location of a single viral particle in the slide. Then the positioning of the laser is more accurate.

## 4. Conclusions

This study provides findings regarding the chikungunya virion diameter (~52 nm), the stiffness of the particle, and the amount of force one viral particle can withstand when subjected to mechanical loads, which is about 25 nN, demonstrating a strong and stiff behavior compared to previous tested viral particles in same conditions. Additionally, a unique fiber formation was found in the viral sample, with lengths over 500 nm and a building block periodicity of around 37 nm. The CHIKV particle demonstrates a plasticity behavior upon mechanical stress and a positive charge distribution around most of its surface, besides some negative spots that are probably related to interactions between the surface proteins. Moreover, the SERS measurements provide viral particle composition and a vibrational signature. In the spectra, it was possible to identify the main bands related to RNA bases, as well as protein-related bands. The finding of the fiber-like structures can shed light on the relationship between CHIKV infection and pathologies related to collagen-rich sites in the body. These results and the analysis of the biomechanical characterization of this virus can aid in better understanding its pathophysiology, adding new data in the description of the physical and structural properties of CHIKV, corroborating models shown in the literature, and providing new insights that can be useful in designing strategies to fight this infectious agent. The findings presented here are of pronounced importance because once we understand these mechanisms, we can find specific drugs or biomolecules that could weaken this viral particle, which is extremely resistant. Moreover, based on the properties observed, such as vibrational modes and adhesion forces, we might be able to understand more about the target sites and the mechanisms of action of the Chikungunya virus.

## Figures and Tables

**Figure 1 viruses-14-02821-f001:**
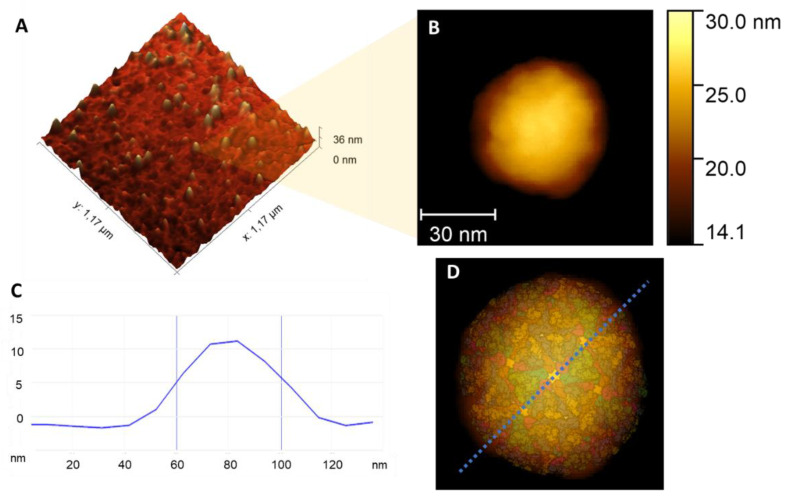
Structure of CHIKV. (**A**) Height image of several viral particles (VPs). The average diameter of the particles is 52.04 nm ± 7.6 nm. (**B**) Height image of an individual CHIKV viral particle. (**C**) Cross-section of a single VP, corresponding to the dotted line in (**D**). (**D**) Correlation between Chikungunya virus modeling [25] and AFM height image.

**Figure 2 viruses-14-02821-f002:**
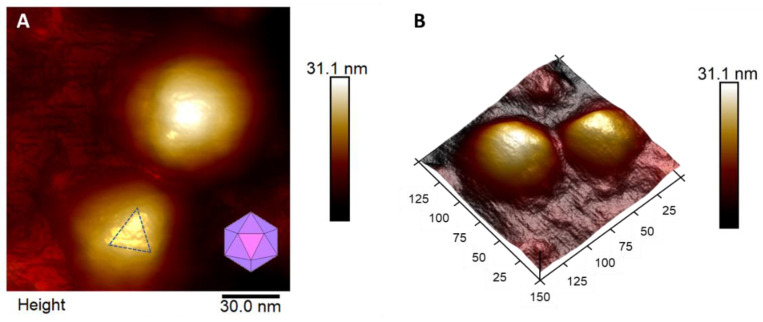
Adsorption patterns. (**A**) Height image of two VPs. Different adsorption patterns evidence the icosahedral geometry of the virus. (**B**) Three-dimensional visualization of image (**A**). The blue dotted triangle highlights the relationship with the inset image, representing virus’ icosahedral geometry.

**Figure 3 viruses-14-02821-f003:**
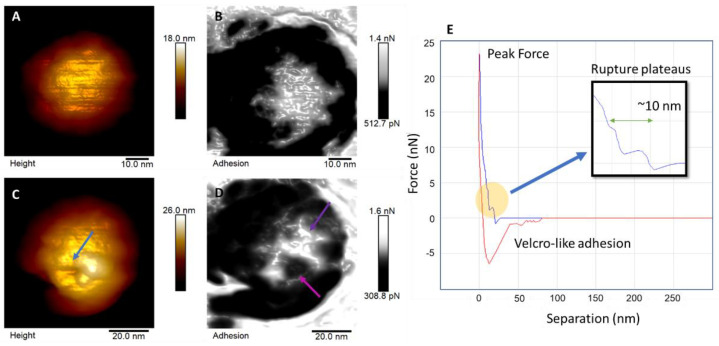
Indentation experiments. (**A**) Height image of a viral particle (VP) and (**B**) its respective adhesion map. At the top of the particle, it is possible to observe differences in charge due to different protein domains. After 30 cycles of indentation, the viral particle shown in image (**C**) reveals the damage caused to the surface of the VP, as evidenced by the blue arrow. It is even possible to observe the pyramidal pattern of the AFM probe geometry. This is associated with the nature of the protein coat of CHIKV, which confers greater rigidity and plasticity. In (**D**), it is possible to observe changes in the distribution of adhesion forces on the VP after indentation, with the purple arrow indicating the site of more positive charge and the pink arrow indicating the site where the indentation occurred, leaving a more negative surface charged location. (**E**) Representative AFM force curve. Force curve showing viral shell rupture (yellow circle). The first event occurs at ~1.4 nN and the second at ~4.1 nN. The insert shows rupture ramps that reveal two layers approximately 10 nm thick (green arrow). The retraction curve (red curve) shows the adhesion force with a Velcro-like pattern, characteristic of protein bond breaks.

**Figure 4 viruses-14-02821-f004:**
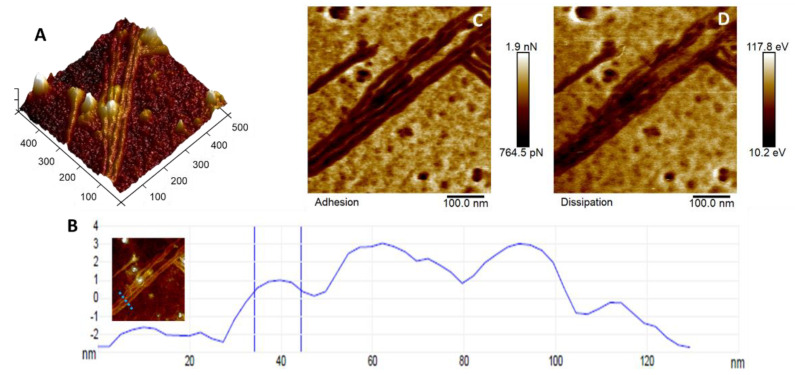
Protein fiber formation: (**A**) 3D height map of the fibers. (**B**) Cross section representing the diameter measurements. The blue dotted line in the inset shows the region from which the cross section was taken. (**C**) Adhesion map demonstrating the contrasts of surface charge of the fibers compared to the substrate, and (**D**) the energy dissipation to assess the viscoelasticity.

**Figure 5 viruses-14-02821-f005:**
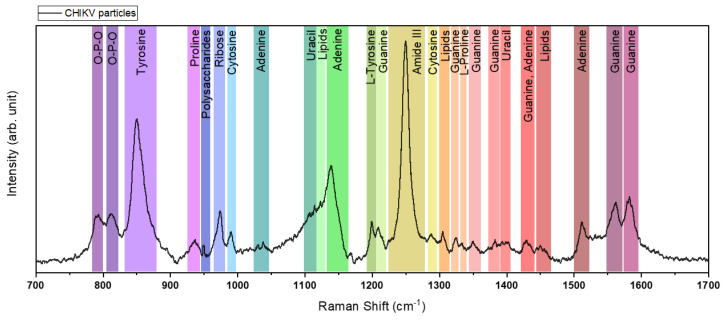
Vibrational signature. Raman spectra were obtained on the solution of viral particles adsorbed on the SERS substrate. Colored bands identify vibrational modes present in the sample.

**Table 1 viruses-14-02821-t001:** Assignments of each band are presented on the SERS spectrum of CHIKV particles.

Wavenumber (cm^−1^)	RNA related	Amino acid/ Protein	Lipid/ Carbohydrate	Reference
793	O-P-O			[42]
812	O-P-O			[43]
851		Tyrosine		[43,44]
837		Proline		[44]
950			Polysaccharides	[42,44]
974	Ribose			[44]
991	Cytosine			[45]
1030	Adenine			[46]
1104	Uracil			[46]
1124			Lipids	[44]
1140	Adenine			[46]
1200		L-Tyrosine		[45]
1210	Guanine			[46]
1230–1260		Amide III		[43]
1288	Cytosine			[47]
1305			Lipids	[48]
1325	Guanine			[43]
1334		L-Proline		[45]
1351	Guanine			[46]
1383	Guanine			[46]
1394	Uracil			[45]
1420–1430	Adenine, Guanine			[43]
1445–1456			Phospholipids, Lipids	[42,44]
1512	Adenine			[49]
1563	Guanine			[46]
1583	Guanine			[43,47]

## Data Availability

The datasets generated during and/or analyzed during the current study are available from the corresponding author upon reasonable request.

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
