# Peer review of "Nanomechanical and Vibrational Signature of Chikungunya Viral Particles"

_viruses, 2022, doi:10.3390/v14122821_

Round 1
Reviewer 1 Report
Dear editor
My considerations follow.
The article is very well designed and written, however some small adjustments are necessary to make the respective work even more attractive when reading.
Abstract: In the summary, make the objective of the work more evident, in addition to better including the material and methods part (detail more), in addition to results and conclusion of the work.
Introduction: At the end of the introduction make it clear what was the objective of the work.
Results and discussion: They are well organized and detailed
Conclusion: The conclusion reflects the findings of the work, however I suggest including more details (be more specific) when mentioning the results given the importance of the findings and what this can reflect for unique health.
Thanks
Author Response
Dear Reviewer, #1 We are thankful for the reviewer's comments on our manuscript. Indeed, the reviewer's suggestions will help improve the manuscript and increase the scientific level. Attached is a point-by-point response to all comments raised by the reviewer.
Reviewer 2 Report
In the manuscript, the authors reported topological, nanomechanical, and vibrational measurements of the Chikungunya Virus via AFM and SERS. Overall, the experimental design is reasonable, and the data is generally clearly presented. However, I did find a couple of issues that need to be addressed.
1. The authors stated in multiple paragraphs that the positive/negative charges of the sample would affect adhesion maps. What is the mechanism? The authors should elaborate on the tip-sample interactions and how they draw conclusions from adhesion results.
2. The authors used indentation curves as an illustration of stiffness, which is insufficient from my perspective. The peak force tapping mode can simultaneously generate modulus and adhesion channels. I think the modulus mapping should also be included, and this will help demonstrate spatial heterogeneity and validate adhesion maps.
3. The authors claimed that the icosahedral geometry of viral particles was observed in figure 2a, which is doubtful. What is the tip radius in the AFM measurement? Is the lateral resolution sufficient to see the sub-structures?
4. In figure 3a and 3b, the center of the particle and the surrounding substrate possesses a higher adhesion. What could be the reason?
5. For the unusual fiber formation, is it due to sample preparation, and is it repeatable? I think the same experiment should be done at least three times to prove the reproducibility.
6. In the SERS spectrum, Amide Ⅰ and lipid peaks at around 1650 cm-1 are missing. Actually, Amide Ⅰ should be the dominant protein peak compared with Amide Ⅲ. The authors need to clarify this peak absence.
7. In the Conclusion section, the authors stated that vibrational signatures (SERS spectra) provide insight into “the protein composition and distribution of viral particles”. This is not true, as SERS does not have the lateral resolution to map protein in single particles.
Author Response
Dear Reviewer, #1
We appreciate the reviewer's comment and would like to thank you in advance for this supportive analysis of our study. Attached is a point-by-point response to all comments raised by the reviewer.

Round 2
Reviewer 2 Report
The authors addressed my questions and concerns in the revised manuscript. I think the manuscript is now good to go.